# OMC-SLIO: Online Multiple Calibrations Spinning LiDAR Inertial Odometry

**DOI:** 10.3390/s23010248

**Published:** 2022-12-26

**Authors:** Shuang Wang, Hua Zhang, Guijin Wang

**Affiliations:** 1Robot Technology Used for Special Environment Key Laboratory of Sichuan Province, Southwest University of Science and Technology, Mianyang 621002, China; 2Shanghai AI Laboratory, Shanghai 200232, China; 3Department of Electrical Engineering, Tsinghua University, Beijing 100089, China

**Keywords:** spinning LiDAR, mechanism calibration, sensor calibration, LiDAR inertial odometry, error-state iterative extended Kalman filter

## Abstract

Light detection and ranging (LiDAR) is often combined with an inertial measurement unit (IMU) to get the LiDAR inertial odometry (LIO) for robot localization and mapping. In order to apply LIO efficiently and non-specialistically, self-calibration LIO is a hot research topic in the related community. Spinning LiDAR (SLiDAR), which uses an additional rotating mechanism to spin a common LiDAR and scan the surrounding environment, achieves a large field of view (FoV) with low cost. Unlike common LiDAR, in addition to the calibration between the IMU and the LiDAR, the self-calibration odometer for SLiDAR must also consider the mechanism calibration between the rotating mechanism and the LiDAR. However, existing self-calibration LIO methods require the LiDAR to be rigidly attached to the IMU and do not take the mechanism calibration into account, which cannot be applied to the SLiDAR. In this paper, we propose firstly a novel self-calibration odometry scheme for SLiDAR, named the online multiple calibration inertial odometer (OMC-SLIO) method, which allows online estimation of multiple extrinsic parameters among the LiDAR, rotating mechanism and IMU, as well as the odometer state. Specially, considering that the rotating and static parts of the motor encoder inside the SLiDAR are rigidly connected to the LiDAR and IMU respectively, we formulate the calibration within the SLiDAR as two separate sets of calibrations: the mechanism calibration between the LiDAR and the rotating part of the motor encoder and the sensor calibration between the static part of the motor encoder and the IMU. Based on such a SLiDAR calibration formulation, we can construct a well-defined kinematic model from the LiDAR to the IMU with the angular information from the motor encoder. Based on the kinematic model, a two-stage motion compensation method is presented to eliminate the point cloud distortion resulting from LiDAR spinning and platform motion. Furthermore, the mechanism and sensor calibration as well as the odometer state are wrapped in a measurement model and estimated via an error-state iterative extended Kalman filter (ESIEKF). Experimental results show that our OMC-SLIO is effective and attains excellent performance.

## 1. Introduction

Given the high reliability and accuracy of light detection and ranging (LiDAR) sensors, LiDAR odometry (LO) and LiDAR simultaneous localization and mapping (SLAM) methods have played an important role in areas such as robotics, autonomous driving, and surveying [1,2,3]. However, the small field of view (FoV) and low vertical resolution of common LiDAR make it unsuitable for specific scenarios, such as unmanned aerial vehicles (UAVs) in indoor scenarios [4,5], autonomous ground vehicles (AGVs) in confined spaces [6,7] or uneven terrain [8,9]. Spinning LiDAR (SLiDAR), which uses an additional rotating mechanism to spin a common LiDAR and scan the surrounding environment, achieves a large FoV at a low cost [10,11]. SLiDARs are widely used in the sensing part of various autonomous robots. For instance, a quadcopter, in which a 2D LiDAR is rotated by the rotor downdraft, can produce a point cloud map [12]. By continuously nodding [11] or spinning a 2D LiDAR [9], the mobile robots can model the surrounding or rough terrain. A 2D turntable equipped with 3D LIDAR, binocular vision and an infrared camera can obtain detailed unstructured terrain information for a hexapod wheel-legged robot [8].

By incorporating an inertial measurement unit (IMU), LiDAR inertial odometry (LIO) can present a better performance for localization and mapping in motion [13]. To make the implementation of LIO efficient and non-professional, self-calibration LIO methods have become a hot research topic in the related community [14,15,16,17]. Unlike the self-calibration odometer for common LiDAR that only considers the calibration between the IMU and the LIDAR, the self-calibration odometer for SLiDAR has to additionally consider the calibration between the rotating mechanism and the LIDAR due to the manufacturing and installation deviation, rotational wear and the nature of thermal expansion and contraction of the rotating mechanism [18]. Therefore, existing self-calibration LIO methods, whether filter-based [14,15] or optimization-based [16,17], do not take the mechanism calibration into account and require the LiDAR to be rigidly attached to the IMU, which cannot be implemented for SLiDAR. In addition, existing mechanism calibration methods, regarding the calibration between the rotating mechanism and the LIDAR, are offline and have very complex implementation principles that hinder their integration to self-calibration LIO [19,20,21].

To address the above issues, we propose firstly a novel self-calibration odometry scheme for SLiDAR, named the online multiple calibration spinning LiDAR inertial odometer (OMC-SLIO) method, which allows online estimation of multiple extrinsic parameters among the LiDAR, rotating mechanism and IMU, as well as the odometer state. Considering the rotating and static parts of the motor encoder inside the SLiDAR are rigidly connected to the LiDAR and IMU, respectively, we formulate the self-calibration within the SLiDAR as two independent sets of calibrations: the mechanism calibration between the LiDAR and the rotating part of the motor encoder and the sensor calibration between the static part of the motor encoder and the IMU. Thus, we construct a well-defined kinematic model from the LiDAR to the IMU with the angular information of the motor encoder. Based on the constructed kinematic model, a two-stage motion compensation method is presented to remove point cloud rotation and motion distortion simultaneously. Finally, we wrap the mechanism calibration, sensor calibration and the odometer state in a measurement model and perform online estimation based on the ESIEKF framework. Experimental results show that our OMC-SLIO is effective and attains excellent performance.

The remainder of this paper is organized as follows. Section 2 discusses related works. Section 3 formulates the problem. Section 4 introduces the proposed OMC-SLIO method in detail. In Section 5, experimental results are presented. We analyze the reason in Section 6. Finally, Section 7 summarizes the work of this paper.

## 2. Related Works

There are plenty of works regarding diverse calibrations among IMUs, cameras and LiDAR, and here we focus on the extrinsic calibration between LiDAR and the IMU and the mechanism calibration of SLiDAR.

### 2.1. Extrinsic Calibration between LiDAR and IMU

There are some tailored offline calibration methods for the LiDAR-IMU system. Ref. [22] uses pre-integration over upsampled IMU measurements derived from a Gaussian process (GP) to remove motion distortion and then combines the factors of IMU pre-integration and LiDAR point-to-plane distances to calibrate extrinsic parameters. A fusion method, in which the ICP and ISPKF are respectively used to determine the unknown transformation and estimate the time delay between the LiDAR and IMU, is presented in [23]. Without any artificial targets or specific facilities, Ref. [24] proposes a multi-feature-based field calibration method for LiDAR-IMU systems that combines point/sphere, line/cylinder and planar features to reduce labor intensity. To associate laser points with stable environmental objects, Ref. [25] adopts a continuous-time IMU trajectory, modeled using GP regression, and segments a point map as structured planes to calibrate LiDAR and an IMU with on-manifold batch optimization. Ref. [26] adopts the B-spline over IMU measurements to formulate a continuous-time trajectory for fusing high-rate scanning points as [25], and uses the point-to-surfel constraint to calibrate parameters. Based on an EKF framework, Ref. [27] utilizes a discrete-time IMU state propagation model to compensate for motion distortion and proposes a motion-based constraint to refine the estimated states. Based on [26,27], Ref. [28] combines Hausdorff distance between the local trajectories and hand-eye calibration to solve the initial spatiotemporal relationship. Then, the IMU pre-integration and the point, line and plane features of the point cloud are wrapped into the objective function. Ref. [29] proposes an informative path planner to find the admissible path for ensuring accurate calibration. However, these offline calibration methods require that the IMU is rigidly attached to LiDAR and cannot be met by SLiDAR.

Regarding online extrinsic calibration in LIO, related works are around the tightly-coupled scheme based on a filter or a nonlinear optimization. Both LINS [14] and FAST-LIO [15] design an error-state iterated Kalman filter (ESIKF) to fuse the IMU and LiDAR. LINS estimates the relative pose between two consecutive local frames for updating the global pose estimate. FAST-LIO nevertheless adopts a scan-to-submap registration with an efficient Kalman gain computation. Besides filter-based methods, nonlinear optimization has been prevailing recently due to its better accuracy. LIOM [16] tightly couples LiDAR and IMU by jointly minimizing their costs in a local fixed-size window and adds an extra rotation constraint to further refine estimation. LIO-SAM [17] also formulates LIO as a factor graph which is akin to LIOM but adopts a keyframe strategy for performing real-time estimation. Although these methods can estimate the extrinsic parameter between the LiDAR and IMU, they also require that the IMU is rigidly attached to LiDAR and does not take mechanism calibration into account.

### 2.2. Mechanism Calibration of SLiDAR

As for the mechanism calibration of SLiDAR, some traditional calibration methods require ad-hoc tools [19,30,31] or regular calibrated scenes [18,32]. There has also been a lot of effort made to relieve calibration in targetless scenes [20,21,33,34,35,36,37].

For calibration in a more general scene, a point-point constraint scheme is utilized [33,34,35]. Based on the random sample consensus (RANSAC) method, Refs. [18,32] extract planes from the whole scene points to construct point-plane constraints for more accurate calibration. Some methods break the calibrated scene into small regions for extracting trivial planes. Refs. [36,37] search for their neighbor points within a suitable radius with the down-sampling scene point cloud. Refs. [20,21] divide the scene into small grids. In each small region, Ref. [20] adopts principal component analysis (PCA) to choose the planes for building point-plane constraints. With the estimated approximated probabilities, Ref. [21] pre-selects the valuable grids to extract planes based on RANSAC. The above methods for SLiDAR mechanism calibration are offline, and their complex principle cannot be directly applied for real-time self-calibration SLIO. In addition, they do not resolve the sensor calibration.

Different from the work in Section 2.1 and Section 2.2, our method can aggregate both the mechanism and sensor extrinsic parameters as well as the odometry state into the ESIEKF framework to realize the online multiple extrinsic calibrations and SLiDAR inertial odometry.

## 3. Problem Formulation

Considering that the rotation and static parts of the encoder are rigidly attached to the LiDAR and IMU, respectively, we can formulate the multiple calibrations inside the SLiDAR platform as two independent sets of calibrations; one is between the LiDAR and the rotating part of the encoder (it exactly corresponds to the rotating mechanism and is therefore named mechanism calibration), and the other is between the static part of encoder and the IMU (named sensor calibration). To illustrate the formulation of multiple calibrations, the internal coordinate relation for SLiDAR is shown in Figure 1.

The subscripts of *L*, *S*, *B* and *I* respectively denote the LiDAR, the rotation part of the encoder, the static part of the encoder and the IMU. The corresponding coordination systems are denoted as ${ℱ}_{L}$, ${ℱ}_{S}$, ${ℱ}_{B}$ and ${ℱ}_{I}$. In addition, the global frame is denoted as ${ℱ}_{G}$. A scanning LiDAR point in ${ℱ}_{L}$, ${ℱ}_{S}$, ${ℱ}_{B}$, ${ℱ}_{I}$ and ${ℱ}_{G}$ is respectively denoted as pL, pS, pB, pI and pG.

Obviously, ${ℱ}_{B}$ and ${ℱ}_{I}$ are rigidly attached to the static part of the SLiDAR platform, and ${ℱ}_{L}$ and ${ℱ}_{S}$ are rigidly attached to the rotation part of the SLiDAR platform. The X coordinates of ${ℱ}_{S}$ and ${ℱ}_{B}$ are collinear, ${ℱ}_{S}$ and ${ℱ}_{B}$ will be exactly coincident, while the spinning angle φ of the encoder is 0 rad. The relative pose TIB between ${ℱ}_{B}$ and ${ℱ}_{I}$ is the sensor calibration denoting the relative pose from the encoder to the IMU, and the relative pose TSL between ${ℱ}_{L}$ and ${ℱ}_{S}$ is the mechanism calibration denoting the relative pose from LiDAR to the mounting point of the rotation mechanism. As a result, TIB and TSL are the interested multiple extrinsic parameters that we need to be estimated online in LIO.

In addition, as our SLiDAR system is unidirectional, mechanism calibration parameters along the spinning direction cannot be estimated due to unobservability [20]. Furthermore, the range bias of LiDAR might be several centimeters, which is much larger than the mechanism bias, so the translation parts of mechanism extrinsic parameters cannot be estimated in a non-specific scene [9]. Given the above reasons, we simplify mechanism calibration as only two rotational parameters ωSL(ωy, ωz) in practice, in which both the rotational parameter in X coordination and all the translational parameters are set as zero. It should be noted that we still use TSL to act as mechanism calibration later for clarity.

## 4. Method

### 4.1. Overall Pipeline

For SLiDAR, the overlap between the consecutive LiDAR scanning frame is minor due to the rapid LiDAR rotation; we therefore adopt a scan-to-submap match approach to ensure enough associations. Meanwhile, built upon an efficient ESIEKF framework [38], we aggregate both mechanism and sensor calibrations as well as odometry poses into one measurement model to pursue real-time performance. The overall pipeline of the proposed OMC-SLIO is shown in Figure 2, which mainly contains four parts: preprocessing, global transformation, ESIEKF and global mapping. The two-stage motion compensation acts as part of the global transformation.

In preprocessing, the planar points are extracted from LiDAR scanning points, the spinning angles are obtained from the encoder, and the IMU states are propagated upon every IMU measurement. In global transformation, extracted planar points are projected to global space via successive transformations consisting of mechanism calibration, motion compensation, sensor calibration and IMU global poses. Specifically, motion compensation covers two kinds of distortions from LiDAR spinning and platform moving. The global planar points are used to associate the global planes in the submap of global mapping. Then, the residuals derived from point-to-plane constraints are passed into ESIEKF to update states. If converged, global mapping and odometry output will be updated, otherwise, global transformation and feature association will be executed again with updated states.

In the following, we will introduce more details on preprocessing, global transformation and ESIEKF; readers can refer to related works for details of global mapping [39].

### 4.2. Preprocessing

The frequencies of the data received from the LiDAR, encoder and IMU are different; for instance, the LiDAR is 10Hz and both the encoder and IMU are 200 Hz in our SLiDAR platform. Thus, before preprocessing, the raw encoder spinning angle, IMU angular rate and IMU acceleration will be temporarily stored in the respective buffers, until receiving one frame of the LiDAR scanning points is completed.

For the raw LiDAR scanning points, the planar points will first be extracted as in LOAM [40], and the extracted planar points will then be projected to the global space. Both the raw spinning angle and IMU data are first utilized in the two-stage motion compensation as introduced in Section 4.3.1. In addition, each angular rate and acceleration of IMU data in the buffer will be propagated to predict the ego-motion states; the detail of state propagation will be introduced in Section 4.4.2.

Note that as the timestamp of the latest rotation angle may be earlier than the timestamp at the end of the current LiDAR scan, this will result in some backward points not being able to be interpolated to correspond to the rotation angle. Therefore, we utilize the accumulated spinning angle measurements to estimate the current spinning rate in the preprocessing. Denote $\omega_{\varphi }^{N-1}$ and $\omega_{\varphi }^{N}$ as the last and current spinning rate, respectively,
(1)$$\omega_{\varphi }^{N}=\frac{\omega_{\varphi }^{N-1}S_{t}^{N-1}+\varphi_{N}}{S_{t}^{N-1}+∆\varphi },\;S_{t}^{N-1}=\sum_{i=1}^{i=N-1}\varphi^{i},\;$$
where ∆φ is the measurement period, $S_{t}^{N-1}$ is the sum of the last *N* − 1 measured spinning angles, and $\varphi_{N}$ is the latest spinning angle. Based on the estimated spinning rate and the time difference, each point in the tail of LiDAR scanning can be integrated into an accurate spinning angle.

### 4.3. Global Transformation

For associating planar features in the submap, we need to project the raw point pL in the LiDAR frame into the global point pG with global transformation. In addition, due to the LiDAR’s continuous spinning and the platform’s moving, all the points received in a scanning period are not in the identity coordination, and thus, we need to transform all the points to the end-time of scanning via motion compensation. According to the coordinate relation of SLiDAR as shown in Figure 1 and considering the motion compensation, the global transformation of SLiDAR contains five consecutive transformations: extrinsic mechanism parameter, LiDAR spinning compensation, platform moving compensation, extrinsic sensor parameter and global IMU pose.

Denote $L_{k-1}$ as the end of the last LiDAR scanning and $L_{k}$ as the end of current LiDAR scanning. Assume pSii is the point in the frame ${ℱ}_{S}$, which is projected from the original pLii in ${ℱ}_{L}$ with the extrinsic mechanism parameter TSL,
(2)pSii=TSLpLii.

Furthermore, denote pSki as the point of end time $L_{k}$ in the frame of the rotation part of the encoder ${ℱ}_{S}$ and $T_{X}\left(\varphi_{s}\right)$ as LiDAR spinning distortion compensation with the corresponding angle $\varphi_{s}$ (derived in Section 4.3.1.1),
(3)pSki=TX(φs)pSii

Then, denote pBii as the point in the frame of the static part of the encoder ${ℱ}_{B}$ and $T_{X}\left(\varphi_{a}\right)$ as the transformation that projects pSki into the frame ${ℱ}_{B}$ with $\varphi_{a}$ angle (derived in Section Removing LiDAR Spinning Distortion of 4.3.1),
(4)pBii=TX(φa)pSki

Furthermore, assume that pBki is the point of end time $L_{k}$ in the frame ${ℱ}_{B}$ and Tki is the distortion compensation of platform moving,
(5)pBki=TkipBii. 

Finally, denote RGI and pGI as the attitude and position of the IMU in the global frame respectively, the corresponding global point pGii is
(6)pGii=RGITIBpBki+pGI. 

In the following, we will derive the details of the two-stage motion compensation, LiDAR spinning compensation and platform moving compensation, respectively.

#### 4.3.1. The Two-Stage Motion Compensation

Different from the common LiDAR odometry, which only resolves the moving distortion, SLiDAR will incur the extra LiDAR spinning distortion. Based on the coordinate relation of SLiDAR in Figure 1, the bottom-up process of the two-stage motion compensation, which removes the LiDAR spinning distortion and platform moving distortion in turn, is shown in Figure 3.

##### 4.3.1.1. Removing LiDAR Spinning Distortion

First, we interpolate the measured spinning angles to get the corresponding spinning angle of the LiDAR point, then rotate the point to the frame ${ℱ}_{B}$ with the relative angle difference.

Assume the sample period of the spinning angle is ∆γ, the end time of current scanning is $\rho_{k}$, the sample time for pSii is $\rho_{i}$, and the spinning rate $\omega_{\varphi }$ is constant during a sample period ∆γ. In Figure 3, $\rho_{i}\in \left[\gamma_{h-1},\gamma_{h}\right]$, in which $\gamma_{h-1}$ and $\gamma_{h}$ are the adjacent time of spinning angle measurement. Denote the measured spinning angles at the time $\gamma_{h-1}$ and $\gamma_{h}$ are $\varphi_{h-1}$ and $\varphi_{h}$, respectively. With the linear interpolation, the spinning angle at the time $\rho_{i}$ is
(7)$$\varphi_{i}=\frac{\gamma_{h}-\rho_{i}}{∆\gamma }\varphi_{h-1}+\frac{\rho_{i}-\gamma_{h-1}}{∆\gamma }\varphi_{h}$$

If $\rho_{i}$ is larger than the latest time $\gamma_{h}$, we can utilize the estimated spinning rate $\omega_{\varphi }$ in time $\gamma_{h}$ to predict the spinning angle corresponding to $\rho_{i}$,
(8)$$\varphi_{i}=\varphi_{h-1}+\left(\rho_{i}-\gamma_{h-1}\right)\omega_{\varphi }$$

Since LiDAR spinning is only along the X-axis, so
(9)$$T_{X}\left(\varphi_{s}\right)=T_{X}\left(\varphi_{i}-\varphi_{k}\right)=\left[R_{X}\left(\varphi_{i}-\varphi_{k}\right),\;0\right]$$

Similarly, we can get the spinning angle $\varphi_{k}$ corresponding to $\rho_{k}$. Denote the measured spinning angle corresponding to the frame ${ℱ}_{B}$ is $\varphi_{0}$, we can derive
(10)$$T_{X}\left(\varphi_{a}\right)=T_{X}\left(\varphi_{k}-\varphi_{0}\right)=\left[R_{X}\left(\varphi_{k}-\varphi_{0}\right),\;0\right]$$

Notably, $R_{X}\left(\varphi \right)$ in both formulations (9) and (10) is defined as:(11)$$R_{X}\left(\varphi \right)=\left[\begin{array}{ccc} 1 & 0 & 0\\ 0 & \cos\varphi & -\sin\varphi \\ 0 & \sin\varphi & \cos\varphi \end{array}\right]$$

From formulations (3), (4), (9) and (10), we can get
(12)pBii=TX(φa)TX(φs)pSii=RX(φk−φ0)RX(φi−φk)pSii=RX(φi−φ0)pSii
which means that we can group $T_{X}\left(\varphi_{s}\right)$ and $T_{X}\left(\varphi_{a}\right)$ as encapsulated LiDAR spinning distortion compensation.

##### 4.3.1.2. Removing Platform Moving Distortion

Regarding platform moving compensation, the back-propagation of IMU measurements is employed to get the related pose for each point, and the point is then transformed to the scanning end-time with its corresponding related pose [15]. Different from the existing LIO methods which regard the point projected by motion compensation as the raw point for the measurement model, since the moving distortion transformation Tki is the middle parameter in the global transformation for SLiDAR, we need to record the corresponding Tki for raw point pLii as part of the measurement model.

Performing the forward propagation on IMU inputs (in Section 4.4.2), we can get a predicted IMU pose chain until the time at the LiDAR scanning end $L_{k}$, in which we denote xˆIGIk as the predicted state. Assume the time $\rho_{i}$ of point pBii located in IMU time interval, that is $\rho_{i}\in \left[t_{j-1},t_{j}\right]$, as shown in Figure 3. By querying the predicted IMU pose chain, we can get the IMU pose xˆIGIj at time $t_{j}$. Then, performing the back-propagation from the initial IMU state xˆIGIj, we can derive the corresponding IMU state xˇIGIi for pBii with the time interval $t_{j}-\rho_{i}$. As for the implementation detail of back-propagation, the interested reader can refer to [15].

We denote the back-propagated IMU poses at the time $\rho_{i}$ as (RˇGIi, pˇGIi )∈xˇIGIi, the front-propagated IMU poses at the end of $L_{k}$ as (RˆGIk, pˆGIk )∈xˆIGIk and the predicted sensor calibration as TˆIB(RˆIB,tˆIB), respectively. As a result, projecting pBii to pBki as
(13)pBki=RˆIBT(RˆGIkT(RˇGIi(RˆIBpBii+tˆIB)+pˇGIi−pˆGIk)−tˆIB)
we can derive moving distortion compensation Tki(Rki,tki) as
(14)Rki=RˆIBTRˆGIkTRˇGIiRˆIB, 
(15)tki=RˆIBT(RˆGIkTRˇGIi−I)tˆIB+RˆIBTRˆGIkT(pˇGIi−pˆGIk). 

For general IMUs, the propagated values pˇGIi and pˆGIk are not reliable, and the related term pˇGIi−pˆGIk in formulation (15) is also usually trivial, so we ignore this term in the initial motion compensation. In the end, estimated odometry results will be used to compensate for distortion again with formulations (14) and (15) while finishing optimization. For pursuing real-time performance, we do not update formulation (14) and (15) with recorrect states during iterative optimization.

### 4.4. ESIEKF

With the ESIEKF optimization framework [38], we aggregate mechanism calibration, sensor calibration and IMU states into one measurement model to build our OMC-SLIO. Specifically, we will introduce the main contents of ESIEKF implementation on OMC-SLIO, which includes the state transition model, forward propagation and ESIEKF update.

#### 4.4.1. State Transition Model

We take the IMU frame ${ℱ}_{I}$ as the body frame and the first body frame of ${ℱ}_{I}$ as the global frame ${ℱ}_{G}$. The true state is defined as
(16)≜[xIGIRIBtIBωSL]T
in which RIB and tIB are respectively the rotation and linear parts of sensor calibration, ωSL=[ωLSyωLSz]T is the interested rotational parameter vector of mechanism calibration as discussed in Section 3, and xIGI denotes the IMU state. xIGI is defined as
(17)xIGI≜[RGIpGIvGIbωbagG]T
where RGI, pGI, vGI and gG denote the IMU attitude, position, velocity and gravity vector in the global frame ${ℱ}_{G}$, respectively, $b_{\omega }$ is the IMU gyroscope bias, and $b_{a}$ is the IMU accelerometer bias. The continuous time kinematic model for state x is:(18)R˙GI=RGI⌊ωm−bω−nω⌋x, p˙GI=vGI,v˙GI=RGI(am−ba−na)+gGb˙ω=nbω,b˙a=nba,g˙G=0,RIB˙=0,t˙IB=0,ω˙SL=0
where $\omega_{m}$ and $a_{m}$ are the raw IMU measurements, $n_{\omega }$ and $n_{a}$ are the measurement white noise of $\omega_{m}$ and $a_{m}$, and both IMU bias $b_{\omega }$ and $b_{a}$ are modeled as random walk and corrupted by Gaussian noise $n_{b\omega }$ and $n_{ba}$. Note that here we only consider two rotation parameters, ωLSy and ωLSz, of mechanism calibration TSL. Denote RSL and tSL as the rotation and translation parts of TSL respectively, then (19)RSL=Rz(ωLSz)Ry(ωLSy),tSL=0,Rz(ωLSz)=[cosωLSzsinωLSz0−sinωLSzcosωLSz0001],Ry(ωLSy)=[cosωLSy0−sinωLSy010sinωLSy0cosωLSy]

With the zero-order hold discretization, the discrete state transition model at the *i*-th IMU measurement can be expressed as [38]:(20)$$x_{i+1}=x_{i}⊞\left(∆tf\left(x_{i},\;u_{i},\;w_{i}\right)\right)$$
where ⊞ is encapsulation operation “boxplus” as defined in [41], ∆t is the IMU sample period, and the discrete kinematics function f, input u, and process noise w are respectively defined as:(21)$$u≜\left[\begin{array}{cc} \omega_{m} & a_{m} \end{array}\right]$$
(22)$$w≜\left[\begin{array}{cc} \begin{array}{cc} n_{\omega } & n_{a} \end{array} & \begin{array}{cc} n_{b\omega } & n_{ba} \end{array} \end{array}\right].$$(23)f(xi,ui,wi)=[ωmi−bωi−nωivGIi+12(RGIi(ami−bai−nai)+gGi)ΔtRGIi(ami−bai−nai)+gGinbωinbai011×1]. 

#### 4.4.2. Forward Propagation

Assume the optimal estimated state of the last LiDAR scanning $L_{k-1}$ at the end time is ${\bar{x}}_{k-1}$ with covariance ${\bar{P}}_{k-1}$, which represents the covariance of the error state defined below:(24)x˜k−1≜xi+1−x¯k−1=[δθGIp˜GIv˜GIb˜ωb˜ag˜GδθIBt˜IBω˜SL]T
where δθGI=Log(R¯GITRGI) is the IMU attitude error, δθIB=Log(R¯IBTRIB) is the error of the rotation part of sensor calibration, and the rests are the standard errors like the form of $\tilde{x}=x-\bar{x}$. Here, δθ vector presents the small deviation between the true and the estimated rotations in the tangent space; it has a minimal representation of three degrees of freedom.

Upon each IMU input $u_{i}$, forward propagation is performed to predict the state ${\hat{x}}_{i+1}$ by setting the process noise $w_{i}$ to zero, and the covariance ${\hat{P}}_{i+1}$ for ${\tilde{x}}_{i+1}≜x_{i+1}⊟{\hat{x}}_{i+1}$(⊟ is also encapsulation operation “boxminus” as defined in [41]) is updated,
(25)$${\hat{x}}_{i+1}={\hat{x}}_{i}⊞\left(∆tf\left({\hat{x}}_{i},\;u_{i},\;0\right)\right),\;{\hat{x}}_{0}={\bar{x}}_{k-1},\;$$
(26)$${\hat{P}}_{i+1}=F_{\tilde{x}}{\hat{P}}_{i}F_{\tilde{x}}^{T}+F_{w}QF_{w}^{T},\;{\hat{P}}_{0}={\bar{P}}_{k-1},\;$$
(27)$$F_{\tilde{x}}={\frac{\partial {\tilde{x}}_{i+1}}{\partial {\tilde{x}}_{i}}|}_{{\tilde{x}}_{i}=0,w_{i}=0\;},\;F_{w}={\frac{\partial {\tilde{x}}_{i+1}}{\partial \;w_{i}}|}_{{\tilde{x}}_{i}=0,w_{i}=0\;}$$
where Q is the covariance of w, and the detailed derivation of $F_{\tilde{x}}$ and $F_{w}$ can be found in [38]. Note that the forward propagation is from time $t_{k-1}$ (the end time of the last LiDAR scanning $L_{k-1}$) to $t_{k}$ (the end time of the current LiDAR scanning $L_{k}$). Denote the propagated state and covariance at $t_{k}$ as ${\hat{x}}_{k}$ and ${\hat{P}}_{k}$, and the error state ${\tilde{x}}_{k}=x_{k}⊟{\hat{x}}_{k}$ complies with the prior distribution,
(28)$${\tilde{x}}_{k}∽N\left(0,\;{\hat{P}}_{k}\right)$$

#### 4.4.3. ESIEKF Update

Denote the current iteration of IEKF as υ, and the corresponding estimated and error states are ${\hat{x}}_{k}^{\upsilon }$ and ${\tilde{x}}_{k}^{\upsilon }$, respectively. In addition, ${\hat{x}}_{k}^{\upsilon }={\hat{x}}_{k}$(if υ=0) and $x_{k}={\hat{x}}_{k}^{\upsilon }⊞{\tilde{x}}_{k}^{\upsilon }$. Define the true planar feature point as
(29)pLigt=pLi−nLi
where nLi is the measurement noise of the LiDAR scanning point. Since pLigt is a planar point, its neighbors in the global map should be on a local plane. By projecting pLigt to the global frame with the true state $x_{k}$, the measurement model can be derived as
(30)hi(xk,nLi)=0=nGiT((RGI(RIB(RkiRX(φi−φ0)RSL(pLi−nLi)+tki)+tIB)+pGI)−cGi)
where RGI∈xk, RIB∈xk, RSL∈xk, tIB∈xk, and pGI∈xk, nGi and cGi are the normal vector and center point of the local plane fitted in the global map.

Ignoring the measurement noise nLi, we can project extracted planar feature point pLi to the global frame with estimated state ${\hat{x}}_{k}^{\upsilon }$,
(31)pGiυ=RˆGIkυ(RˆIBυ(RkiRX(φi−φ0)RˆSLυpLi+tki)+tˆIBυ)+pˆGIkυ
where RˆGIkυ∈xˆkυ, RˆIBυ∈xˆkυ, RˆSLυ∈xˆkυ, tˆIBυ∈xˆkυ and pˆGIkυ∈xˆkυ. Similarly, since pLi is a planar point, its neighbors in the global map should be on a local plane, so we can define
(32)hi(xˆkυ,0)=nGiT(pGiυ−cGi)=0
where nGi and cGi are normal vector and center point of the local plane fitted by the neighboring points of pGiυ in the global map. As a result, we can define the residual as
(33)riυ≜hi(xk,nLi)⊟hi(xˆkυ,0)=hi(xˆkυ⊞x˜kυ,nLi)⊟hi(xˆkυ,0)≈Hiυx˜kυ+DiυnLi, 
(34)Hiυ=δhi(xˆkυ⊞x˜kυ,0)δx˜kυ|x˜kυ=0, Diυ=δhi(xˆkυ,nLi)δnLi|nLi=0, 
where the detailed derivation for $H_{i}^{\upsilon }$ and $D_{i}^{\upsilon }$ can be referred to in [38], and the formulation (33) defines a posteriori distribution for ${\tilde{x}}_{k}$,
(35)(DiυnLi)|x˜kυ=riυ−Hiυx˜kυ~N(0,R¯i), R¯i=DiυRiDiυT. 

After every iteration, the prior distribution of ${\tilde{x}}_{k}$ from the forward propagation has been evolved as:(36)$${\tilde{x}}_{k}=x_{\kappa }⊟{\hat{x}}_{\kappa }=\left({\hat{x}}_{k}^{\upsilon }⊞{\tilde{x}}_{k}^{\upsilon }\right)⊟{\hat{x}}_{\kappa }\approx {\hat{x}}_{k}^{\upsilon }⊟{\hat{x}}_{\kappa }+J^{\upsilon }{\tilde{x}}_{k}^{\upsilon }$$
(37)$$J^{\upsilon }={\left(\frac{\partial \left(\left({\hat{x}}_{k}^{\upsilon }⊞{\tilde{x}}_{k}^{\upsilon }\right)⊟{\hat{x}}_{\kappa }\right)}{\partial {\tilde{x}}_{k}^{\upsilon }}\right)}_{{\tilde{x}}_{k}^{\upsilon }=0}$$
where the detailed derivation for $J^{\upsilon }$ can be found in [38], and ${\hat{x}}_{k}^{\upsilon }={\hat{x}}_{\kappa }$, $J^{\upsilon }=I$ while υ=0. From formulation (36), we can find the evolutive prior distribution for ${\tilde{x}}_{k}^{\upsilon }$ is,
(38)$${\tilde{x}}_{k}^{\upsilon }∽N\left(-J^{\upsilon }{}^{-1}\left({\hat{x}}_{k}^{\upsilon }⊟{\hat{x}}_{\kappa }\right),\;J^{\upsilon }{}^{-1}{\hat{P}}_{k}J^{\upsilon }{}^{-T}\right).$$

Furthermore, combine the prior in formulation (38) and the posteriori distribution in formulation (35), which leads to the maximum a-posteriori estimate (MAP) for ${\tilde{x}}_{k}^{\upsilon }$,
(39)$$\underset{{\tilde{x}}_{k}^{\upsilon }}{\min}\left(||{\hat{x}}_{k}^{\upsilon }⊟{\hat{x}}_{\kappa }+J^{\upsilon }{\tilde{x}}_{k}^{\upsilon }||{}_{{\hat{P}}_{k}{}^{-1/2}}^{2}+\sum ||r_{i}^{\upsilon }-H_{i}^{\upsilon }{\tilde{x}}_{k}^{\upsilon }{||}_{{\bar{R}}_{i}^{-1/2}}^{2}\right),\;$$
where ${||x||}_{\Sigma }^{2}=x^{T}\Sigma^{T}\Sigma x$. Substituting the linearization of the priori in the above equation and optimizing the resultant quadratic cost leads to the standard iterated Kalman filter,
(40)$$K=PH^{T}{\left(HPH^{T}+\bar{R}\right)}^{-1},$$
(41)$${\hat{x}}_{k}^{\upsilon +1}={\hat{x}}_{k}^{\upsilon }⊞\left(Kr^{\upsilon }+\left(I-KH\right){\left(J^{\upsilon }\right)}^{-1}\left({\hat{x}}_{k}^{\upsilon }⊟{\hat{x}}_{k}\right)\right),$$
where $H={\left[H_{1}^{\upsilon }{}^{T},\cdots ,H_{m}^{\upsilon }{}^{T}\right]}^{T}$, $\bar{R}=diag\left({\bar{R}}_{1},\cdots ,{\bar{R}}_{m}\right)$, $P={\left(J^{\upsilon }\right)}^{-1}{\hat{P}}_{k}{\left(J^{\upsilon }\right)}^{-T}$, and $r^{\upsilon }={\left[r_{1}^{\upsilon }{}^{T},\cdots ,r_{m}^{\upsilon }{}^{T}\right]}^{T}$. The updated estimate ${\hat{x}}_{k}^{\upsilon +1}$ is then used to compute the residual again and the process is repeated until convergence. After convergence, the optimal state estimation and covariance is
(42)$${\bar{x}}_{k}={\hat{x}}_{k}^{\upsilon +1},\;{\bar{P}}_{k}=\left(I-KH\right)P.$$

## 5. Experiment Results

In order to verify the proposed OMC-SLIO, we designed a SLiDAR experimental platform, as shown in Figure 4, in which the hardware system can be divided into two parts: the rotating part and the static part. Specifically, the rotating part consists of Velodyne VLP-16 LiDAR, the rotating mechanism and the rotating part of a DJI GM6020 motor (including the rotating part of the embedded encoder); the static part consists of the static part of a DJI GM6020 motor (including the static part of the embedded encoder), a XSENS MTi-630 IMU, a Piesia WK310CA computer (Intel Core i78565u CPU and 16 GB RAM) and a metal square base. Note that the Velodyne VLP-16 LiDAR has been mounted on the rotating mechanism as accurately as possible to ensure that all external parameters of the mechanism such as ωLSy and ωLSz are approaching zero.

We chose the state-of-the-art (SOTA) self-calibration methods, FAST-LIO [15] and LIO-SAM [17], and the loosely coupled method, LOAM [40], as comparison methods. Since the relative pose between LiDAR and an IMU is non-constant and severe point cloud distortion results from LiDAR spinning, these comparison methods would lead to failure when they are directly applied for SLiDAR. Therefore, based on our constructed kinematic model and the two-step point cloud distortion compensation method, we transformed the original LiDAR scanning points to the space of the static part of the encoder (i.e., ${ℱ}_{B}$); then, the newly formed LiDAR scanning points were fed to these comparison methods. To distinguish the original forms of these comparison methods, we term these modified FAST-LIO, LIO-SAM and LOAM for SLiDAR as FAST-SLIO, SLIO-SAM and SLOAM, respectively.

Furthermore, to validate the correctness of our formulated SLiDAR calibration, we adopted two specific offline calibration methods to estimate the mechanism external parameters [27] and sensor external parameters [21], respectively. Then, the online calibration parameters in OMC-SLIO were replaced with the corresponding external parameters from the above offline calibration. Again, we refer to such pre-calibrated OMC-SLIO as the Calib-SLIO.

We conducted two sets of comparison experiments. One set of comparison experiments was conducted in a lab room with a Nokov Motion Capture System (NMCS) (shown in Figure 5), where the positioning results of the motion capture system were used as ground truth. Another set of comparison experiments was conducted in a larger underground parking to reveal the effect of external parameter self-calibration on localization and mapping.

### 5.1. Lab Room

In the lab room test, the positioning result from the NMCS is regarded as the ground truth. The absolute translation errors (ATE) presented by MAE and RMSE and the average processing time (APT) for each LiDAR scanning (10 Hz) are listed in Table 1. The mapping results of the lab room using the comparison methods are shown in Figure 6. Note that we only present the results of comparative methods that worked successfully.

In Table 1, it can be found that our OMC-SLIO, Calib-SLIO and FAST-SLIO all achieve centimeter-level localization accuracy, while both SLOAM and SLIO-SAM fail to function successfully. Since fast LiDAR spinning presents minor overlap between successive frame scanning, the failure of SLOAM and SLIO-SAM stems from their basic matching based on successive frames. Further, our OMC-SLIO, pre-calibrated Calib-SLIO and FAST-SLIO pre-processed using our proposed method all achieve reliable localization and mapping, indicating the effectiveness of our proposed SLiDAR calibration formulation and the two-step motion compensation.

Specifically, our OMC-SLIO shows a significant improvement in Z-directional localization, with a 40% improvement in accuracy compared to FAST-SLIO. Furthermore, the Z-direction localization of all methods exhibited the largest MAE and RMSE values, owing to the weaker laser beam reflection intensity from the smooth ground (as shown in Figure 6, the green and red points indicate strong and weak laser beam reflections, respectively). For the APT metric, the average runtime of our OMC-SLIO, Calib-SLIO and FAST-SLIO is around 35 ms. This APT result is reliable because our OMC-SLIO, Calib-SLIO, and FAST-SLIO are all able to achieve similar convergence in the lab room. Consistent with quantitative results, our OMC-SLIO shows sharper mapping details than FAST-SLIO and presents comparable mapping performance with Calib-SLIO, as shown in Figure 6.

### 5.2. Underground Parking

In the large underground parking, we cannot use the motion capture system and cannot receive GPS positioning signals. Since Calib-SLIO got the best positioning accuracy in the lab room experiments, the positioning results of Calib-SLIO were adopted as the ground truth in the underground parking test. The absolute translation errors (ATE) presented via MAE and RMSE and the average processing time (APT) for each LiDAR scanning (10 Hz) are presented in Table 2. The underground parking mapping results for the comparison method are shown in Figure 7. Again, we only present the results for the methods that worked successfully.

There is no doubt that SLOAM and SLIO-SAM still fail when applied to SLiDAR. Specifically, compared with FAST-SLIO, our OMC-SLIO has an average improvement of 50% in positioning accuracy in the X and Y directions and 35% in the Z direction, as shown in Table 2. Similarly, since the dark-colored floor and roof can absorb most of the laser beam energy (as shown in Figure 7, green and red dots indicate strong and weak laser beam reflections, respectively), both MAE and RMSE values in the Z direction in Table 2 are worse than those in the X or Y directions. Meanwhile, OMC-SLIO also has better real-time performance than FAST-SLIO due to better estimation convergence. Since the estimation bias of the rotation part of the extrinsic calibration will lead to more severe point cloud bias in larger scenes, the FAST-SLIO experiments in the underground parking exhibit more prominent localization and mapping errors than the experiments in the lab room. Consistent with quantitative results, OMC-SLIO exhibits clearer mapping details, while FAST-SLIO presents a distinctly blurred mapping, as shown on the right in Figure 7.

## 6. Discussion

In this section, we discuss why our OMC-SLIO can show better performance for SLiDAR compared to the comparative self-calibration odometer method, as shown in the previous experimental section.

The best positioning accuracy using Calib-SLIO verified the practicability of the calibration formulation within the SLiDAR. To some extent, our OMC-SLIO can be considered as an online version of Calib-SLIO. Our OMC-SLIO online calculates both mechanism and sensor parameters within SLiDAR, thus achieving excellent positioning and mapping accuracy. In the set of underground parking experiments, we used the offline estimated extrinsic parameters of Calib-SLIO as the ground truth. Then, all the extrinsic parameters estimated online using FAST-SLIO and OMC-SLIO were averaged and taken as the final extrinsic parameter results. Table 3 shows the errors in the online extrinsic parameter estimation of FAST-SLIO and OMC-SLIO.

Note that FAST-SLIO estimates only the extrinsic parameters of the sensor, while our OMC-SLIO estimates both the extrinsic parameters of the mechanism and the sensor. As shown in Table 3, compared with FAST-SLIO, OMC-SLIO reduces the online calibration errors by 43%, 44%, 26%, 72%, 27% and 91% on average for the 6 sensor extrinsic parameters, respectively. Meanwhile, the average error of the two additional online mechanism calibration parameters of OMC-SLIO is only 0.057° and 0.218°. FAST-SLIO does not consider the correlation of the extrinsic parameters inside the SLiDAR, while our OMC-SLIO establishes the kinematic model relationship between multiple sensors inside the SLiDAR and, on the other hand, constructs a unified observation model with multiple extrinsic parameters and the odometry states. Therefore, OMC-SLIO can be more reliable for SLiDAR to achieve better localization and mapping performance.

## 7. Conclusions

In this paper, we propose a novel self-calibration odometry scheme for SLiDAR, named as online multiple calibration spinning LiDAR inertial odometer (OMC-SLIO), which allows online estimation of multiple extrinsic parameters among the LiDAR, rotating mechanism and IMU, as well as the odometer state. To the best of our knowledge, this is the first multi-extrinsic self-calibration LIO method for SLiDAR. By analyzing the relationships of the internal sensors, we formulate the calibration inside the SLiDAR as mechanism calibration and sensor calibration and build the kinematic model with the encoder information. Further, based on the kinematic model, we present a two-stage motion compensation to eliminate the point cloud distortion caused by LiDAR rotation and platform motion. Finally, the two sets of extrinsic parameters and odometer state are wrapped in a measurement model and estimated with the error-state iterative extended Kalman filter (ESIEKF). We design a 3D SLiDAR platform and conduct experiments in a lab room and underground parking. Experimental results show that our OMC-SLIO is effective and attains excellent performance. Further, the extrinsic parameters comparison test verifies that our OMC-SLIO can better estimate the multiple external parameters of SLiDAR and shows why our OMC-SLIO can achieve higher accuracy in localization and mapping. To maintain the consistence of the localization and mapping of our proposed OMC-SLIO, we will introduce the pose graph optimization resulted from the multiple frames match or the loop closure in future work.

## Figures and Tables

**Figure 1 sensors-23-00248-f001:**
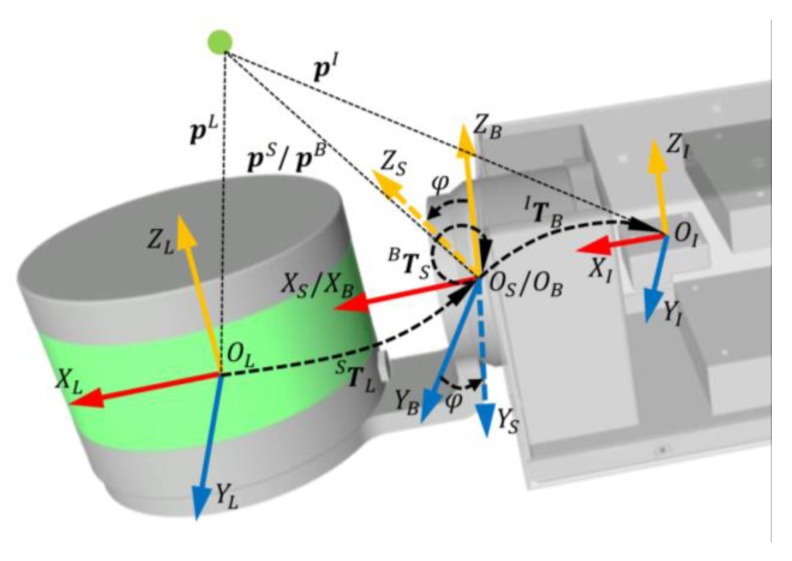
Internal coordinate relation for SLiDAR.

**Figure 2 sensors-23-00248-f002:**
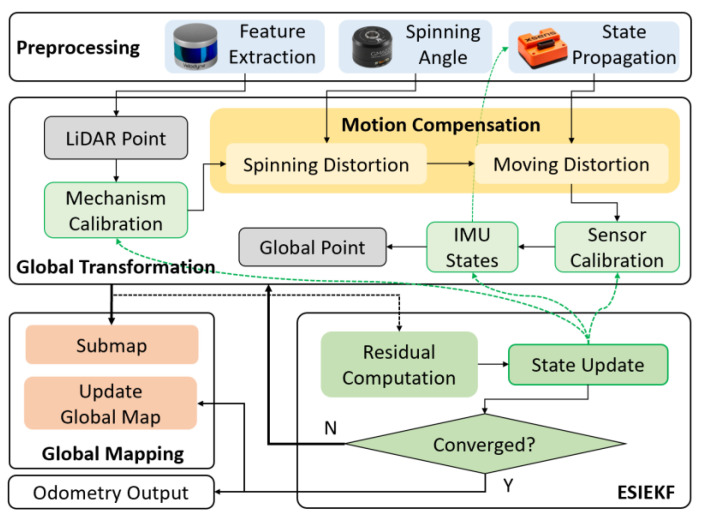
The overall pipeline of proposed OMC-SLIO.

**Figure 3 sensors-23-00248-f003:**
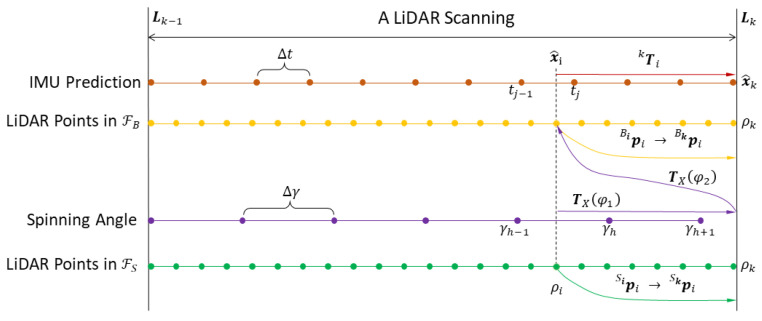
The two-stage motion compensation. The green and yellow dots represent the sequence of LiDAR scan points before LiDAR spinning and platform moving compensation respectively. The purple dots represent the measured spinning angle sequence. The brown dots represent the sequence of measured IMU data.

**Figure 4 sensors-23-00248-f004:**
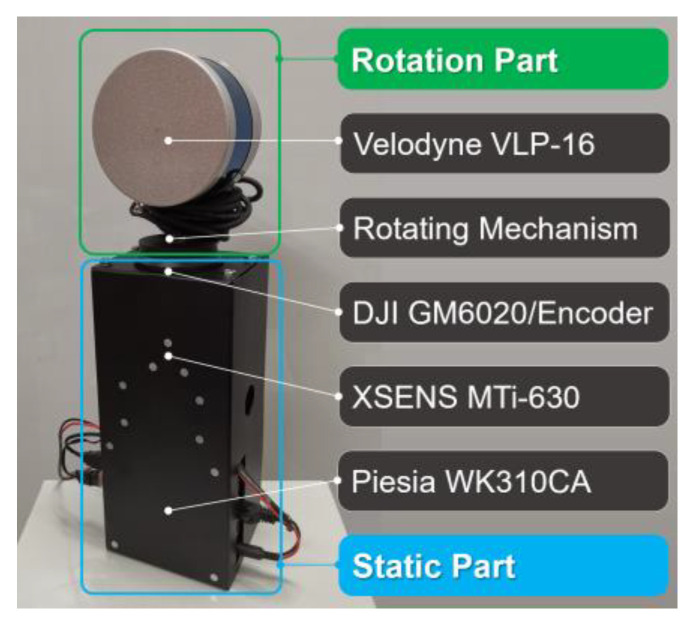
3D SLiDAR experimental platform.

**Figure 5 sensors-23-00248-f005:**
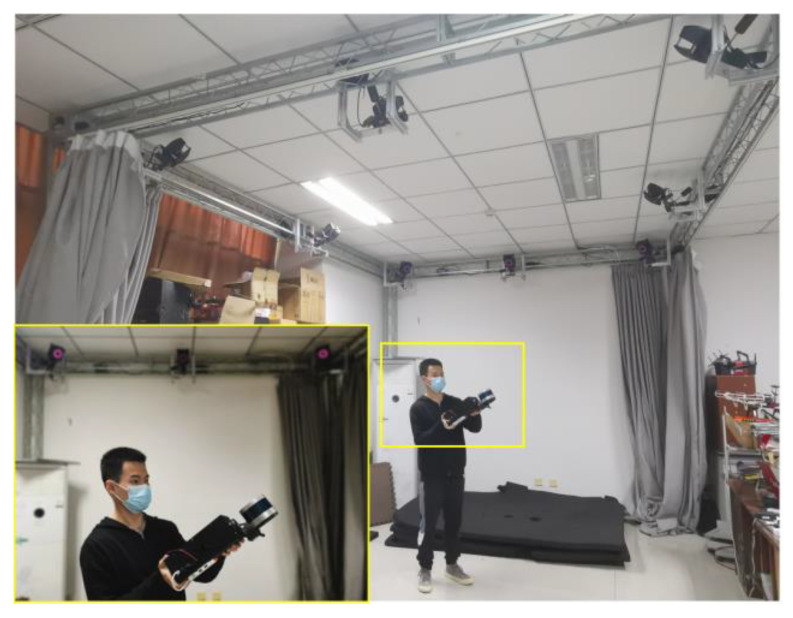
Lab room with a Nokov Motion Capture System.

**Figure 6 sensors-23-00248-f006:**
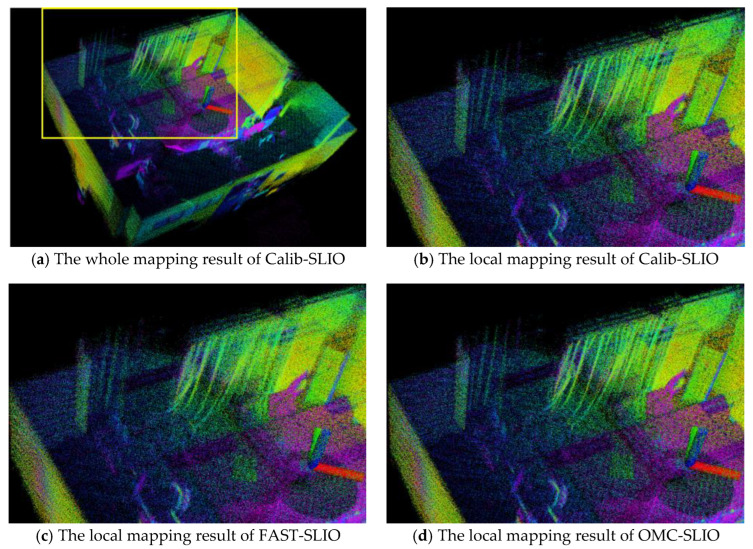
The mapping results of the lab room with Calib-SLIO, FAST-SLIO and OMC-SLIO. Note that the green and red dots show the strong and weak laser beam reflection intensities, respectively.

**Figure 7 sensors-23-00248-f007:**
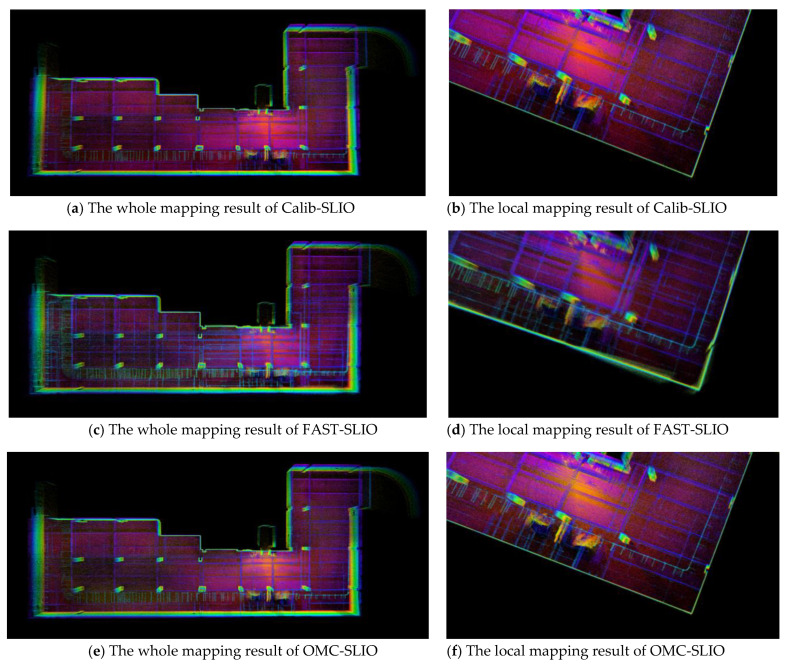
The mapping results of underground parking with Calib-SLIO, FAST-SLIO and OMC-SLIO methods. Note that the green and red dots show the strong and weak laser beam reflection intensities, respectively.

**Table 1 sensors-23-00248-t001:** The ATE and APT of the methods in the lab room.

Method	X-Y-Z	Calib-SLIO	SLOAM	SLIO-SAM	FAST-SLIO	OMC-SLIO
MAE (cm)	X	2.41		Failed	4.05	3.58
	Y	2.92	Failed		4.03	3.69
	Z	4.87			8.59	5.19
RMSE (cm)	X	4.59		Failed	5.24	5.93
	Y	4.23	Failed		5.26	5.91
	Z	5.13			9.28	5.53
APT (ms)	-	34.43	Failed	Failed	34.72	35.49

Ground truth: From Nokov Motion Capture System.

**Table 2 sensors-23-00248-t002:** The ATE and APT of the methods in underground parking.

Method	X-Y-Z	SLOAM	SLIO-SAM	FAST-SLIO	OMC-SLIO
MAE (cm)	X		Failed	10.80	5.47
	Y	Failed		15.86	7.04
	Z			25.29	15.19
RMSE (cm)	X		Failed	12.28	6.48
	Y	Failed		16.64	8.98
	Z			30.19	20.36
Time (ms)	-	Failed	Failed	67.50	49.32

Ground truth: from Calib-SLIO.

**Table 3 sensors-23-00248-t003:** The error of online extrinsic parameter estimation of FAST-SLIO and OMC-SLIO.

Calibration		MAE		RMSE	
		FAST-SLIO	OMC-SLIO	FAST-SLIO	OMC-SLIO
Mechanism	ry (deg)	-	0.056	-	0.058
	rz (deg)	-	0.213	-	0.222
Sensor	rx (deg)	0.269	0.151	0.303	0.172
	ry (deg)	0.652	0.334	0.815	0.488
	rz (deg)	0.611	0.454	0.650	0.491
	tx (cm)	2.3	0.7	2.6	0.7
	ty (cm)	1.8	1.3	1.9	1.4
	tz (cm)	10.4	0.9	11.6	1.1

## Data Availability

The data presented in this study is not publicly available at this time but may be obtained from the authors upon reasonable request.

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
