# Peer review of "OMC-SLIO: Online Multiple Calibrations Spinning LiDAR Inertial Odometry"

_sensors, 2022, doi:10.3390/s23010248_

Round 1
Reviewer 1 Report
In this paper, an online multiple calibrations spinning LiDAR inertial odometry (OMC-SLIO) method is proposed. The innovation points of this article are not clearly described and need major modifications. The details are as follows:
1. First of all, the main contribution of this paper is insufficient. In order to strengthen the motivation, it is better to discuss the details with contributions point by point and should be compared with the latest related works.
2. In "Introduction" section Related Works, I feel the current coverage of the state of the art is not satisfactory as the related work section does not cover many contributions that likely provide the building blocks of the proposed approach. For example,
(1) Flexible gait transition for six wheel-legged robot with unstructured terrains
It is suggested to cite the above articles and analyze the differences in Section Related Works.
3. There are some mistakes including grammar, words and English expression in this paper. Please check the overall paper carefully.
4. Both "Abstract" and "Introduction" sections of the manuscript are not well organized. The "Abstract" section can be made much more impressive by highlighting your contributions.
5. Discussion section should be added and the results should be discussed in detail. The Discussion section should be edited in a more highlighting, argumentative way. The author should analyze the reason why the tested results are achieved. The performance of the proposed method should be better analyzed, commented and visualized in the experimental section.
After the above problems are solved, the manuscript can be considered to be accepted.
Author Response
Dear Editor,
Thank you for kind work of our manuscript (an updated version of previous manuscript: “sensors-2020902 OMC-SLIO: Online Multiple Calibrations Spinning LiDAR Inertial Odometry”), with an opportunity to address the reviewers’ concerns. Based on the comments of the reviewers, we have made corresponding changes noticeable in updated manuscript and answered all concerned questions in this letter.
We are uploading our point-by-point response to the comments (below) and an updated manuscript with changes highlighted.
Best wishes,
Guijin Wang, et al.
Thanks very much for reviewer’s comments. All changes mentioned below are green highlighted in the updated manuscript.
Response to Reviewer 1:
Concern 1: First of all, the main contribution of this paper is insufficient. In order to strengthen the motivation, it is better to discuss the details with contributions point by point and should be compared with the latest related works.
Author response: Thanks for the reviewer’s recommendation. We regret that the contribution was not clearly expressed in the previous manuscript. In the updated manuscript, we highlight the research point of this paper in the abstract and introduction part: self-calibration SLiDAR inertial odometry. In the section of Introduction, we summarize the shortcomings of existing methods. Then, we condense several contributions of the proposed method:
- Considering that the rotating and static parts of the motor encoder inside the SLiDAR are rigidly connected to the LiDAR and IMU, respectively, we formulate the calibration within the SLiDAR as two separate sets of calibrations: the mechanism calibration between the LiDAR and the rotating part of the motor encoder, and the sensor calibration between the static part of the motor encoder and the IMU.
- Based on such SLiDAR calibration formulation, we can construct a well-defined kinematic model from LiDAR to IMU with the angular information from the motor encoder.
- Based on the kinematic model, a two-stage motion compensation method is presented to eliminate the point cloud distortion resulted from LiDAR spinning and platform motion. Furthermore, the mechanism and sensor calibration as well as the odometer state are wrapped in a measurement model and estimated by an Error-state Iterative Extended Kalman Filter (ESIEKF).
We have optimized the description of the contributions in abstract and introduction, and also updated some of the latest work related to SLiDAR for robotic applications and LiDAR-IMU calibration.
Concern 2: In "Introduction" section Relate Works, I feel the current coverage of the state of the art is not satisfactory as the related work section does not cover many contributions that likely provide the building blocks of the proposed approach. For example,
(1) Flexible gait transition for six wheel-legged robot with unstructured terrains
It is suggested to cite the above articles and analyze the differences in Section Related Works.
Author response: Thanks for the reviewer’s advice. We have surveyed and studied the articles suggested by reviewer and related works others. These articles also gave us some inspiration, because they all the application of SLiDAR in different types of robots and different scenarios.
At first, we add some references related to SLiDAR applications in the Introduction section, like
“SLiDAR are widely used in the sensing part of various autonomous robots. For instance, a quadcopter, in which a 2D LiDAR is rotated by the rotor downdraft, can produce a point cloud map [2]. By continuously nodding [5] or spinning a 2D LiDAR [4], the mobile robots can model the surrounding or the rough terrain. A 2D turntable equipped with 3D LIDAR, binocular vision and infrared camera can obtains detailed unstructured terrain information for a hexapod wheel-legged robot [1]”.
Further, we add more related works in Section Related Works.
Concern 3: There are some mistakes including grammar, words, and English expression in this paper. Please check the overall paper carefully.
Author response: Thanks for pointing out the mistakes. We have tried our best to check the grammar, words, and expressions of the article in the updated manuscript.
Concern 4: Both "Abstract" and "Introduction" sections of the manuscript are not well organized. The "Abstract" section can be made much more impressive by highlighting your contributions.
Author response: Thanks for the reviewer’s recommendation. We have reorganized and re-written the Abstract and Introduction sections, where the contribution points of the article are highlighted and clarified.
Concern 5: Discussion section should be added and the results should be discussed in detail. The Discussion section should be edited in a more highlighting, argumentative way. The author should analyze the reason why the tested results are achieved. The performance of the proposed method should be better analyzed, commented, and visualized in the experimental section.
Author response: Thanks for the reviewer’s suggestion. In the previous manuscript, we conducted two sets of localization mapping experiments and extrinsic parameter estimation experiments together. The two sets of localization mapping experiments directly demonstrate the advantages of our proposed method, while the extrinsic parameter estimation experiments further reveal the reasons why our proposed method is better. Therefore, in order to arrange the experiment results more rationally, in the updated manuscript we re-arrange the extrinsic parameter estimation experiments into Discussion and give more detailed description of the reasons why the proposed method can get better results.
Reviewer 2 Report
The approach for an online multiple calibrations spinning LiDAR inertial 471 odometry (OMC-SLIO) method
is interesting and the presentation of the theoretical aspects are very good.
The mathematical model is well founded and adequate to this method. It would be good if the author would present more advantages for this method. It would also be interesting to see the advantages regarding the an online multiple calibrations spinning LiDAR inertial 471 odometry (OMC-SLIO) method
in comparison with the similar methods.
The conclusions could be more detailed: maybe a comparison between the experimental results and the simulation results
Author Response
Dear Editor,
Thank you for kind work of our manuscript (an updated version of previous manuscript: “sensors-2020902 OMC-SLIO: Online Multiple Calibrations Spinning LiDAR Inertial Odometry”), with an opportunity to address the reviewers’ concerns. Based on the comments of the reviewers, we have made corresponding changes noticeable in updated manuscript and answered all concerned questions in this letter.
We are uploading our point-by-point response to the comments (below) and an updated manuscript with changes highlighted.
Best wishes,
Guijin Wang, et al.
Thanks very much for reviewer’s comments. All changes mentioned below are green highlighted in the updated manuscript.
Response to Reviewer 2:
Concern 1: The mathematical model is well founded and adequate to this method. It would be good if the author would present more advantages for this method. It would also be interesting to see the advantages regarding the an online multiple calibrations spinning LiDAR inertial 471 odometry (OMC-SLIO) method in comparison with the similar methods.
Author response: Thanks for reviewer’s recognition and recommendation. As stated in abstract, “Unlike common LiDAR, in addition to the calibration between the IMU and the LiDAR, self-calibration odometer for SLiDAR must also consider the mechanism calibration between the rotating mechanism and the LiDAR. However, existing self-calibration LIO methods require the LiDAR to be rigidly attached to the IMU and do not take the mechanism calibration into account, which cannot be applied to the SLiDAR.” Our proposed method OMC-SLIO is the first self-calibration odometry for SLiDAR.
In the section of Experiment Results, we still choose some current SOTA LiDAR odometry methods, which consist of loosely coupled inertial odometry and tightly coupled self-calibration inertial odometry. These comparison methods still need to be pre-processed with our proposed calibration formulation and motion compensation for applying to SLiDAR. Thus in the updated manuscript, these modified FAST-LIO, LIO-SAM, and LOAM for SLiDAR are renamed as FAST-SLIO, SLIO-SAM, and SLOAM.
Concern 2: The conclusions could be more detailed: maybe a comparison between the experimental results and the simulation results.
Author response: Thanks for the reviewer’s recommendation. In the updated manuscript, we have added a Discussion section to discuss in detail the reasons why our proposed method yields better results, and have further refined the Conclusion section.
Reviewer 3 Report
1. In the introduction, there is very little introduction about the spinning LiDAR calibration method is introduced, please add more related contents and literatures.
2. Please add the contrast test between this paper's methods and other method too verify the validity of this paper's method.
Author Response
Dear Editor,
Thank you for kind work of our manuscript (an updated version of previous manuscript: “sensors-2020902 OMC-SLIO: Online Multiple Calibrations Spinning LiDAR Inertial Odometry”), with an opportunity to address the reviewers’ concerns. Based on the comments of the reviewers, we have made corresponding changes noticeable in updated manuscript and answered all concerned questions in this letter.
We are uploading our point-by-point response to the comments (below) and an updated manuscript with changes highlighted.
Best wishes,
Guijin Wang, et al.
Thanks very much for reviewer’s comments. All changes mentioned below are green highlighted in the updated manuscript.
Response to Reviewer 3:
Concern 1: In the introduction, there is very little introduction about the spinning LiDAR calibration method is introduced, please add more related contents and literatures.
Author response: Thanks for the reviewer’s recommendation. In the Introduction section, we summarize the work on SLiDAR calibration only, e.g.
“In addition, existing mechanism calibration methods regarding the calibration between the rotating mechanism and the LIDAR are offline and have very complex implementation principles that hinder their integration to self-calibration LIO [19-21].”
According to the organization of the manuscript, more details about the SLiDAR calibration work are presented accordingly in the Related Works section.
Concern 2: Please add the contrast test between this paper's methods and other method to verify the validity of this paper's method.
Author response: Thanks for the reviewer’s recommendation. To response this concern, we re-wrote the section of Experiment Results.
As stated in Experiment Results section, “We choose the state-of-the-art self-calibration methods, FAST-LIO [15] and LIO-SAM [17], and the loosely coupled method, LOAM [40], as comparison methods. Since the relative pose between LiDAR and IMU is non-constant and the severe point cloud distortion resulted from LiDAR spinning, these comparison methods will lead to failure when they are directly applied for SLiDAR. Therefore, based on our constructed kinematic model and the two-step point cloud distortion compensation method, we first transform the original LiDAR scan points to the space of the static part of the encoder (i.e. ); then, the newly formed LiDAR scan points will be fed to these comparison methods. To distinguish the original forms of these comparison methods, we term these modified FAST-LIO, LIO-SAM, and LOAM for SLiDAR as FAST-SLIO, SLIO-SAM, and SLOAM, respectively”,
“Furthermore, to validate the correctness of our formulated SLiDAR calibration, we adopted two specific offline calibration methods to estimate the mechanism external parameters [27] and sensor external parameters [21], respectively. Then, the online calibration parameters in OMC-SLIO are replaced with the corresponding external parameters from above offline calibration. Again, we refer to such pre-calibrated OMC-SLIO as the Calib-SLIO”,
and “We conduct two sets of comparison experiments. One set of comparison experiment is conducted in a lab room with a Nokov Motion Capture System (NMCS) (shown in Figure 5), where the positioning results of the motion capture system are be used as ground truth. Another set of comparison experiment is conducted in a larger underground parking to reveal the effect of external parameter self-calibration on localization and mapping”.
Reviewer 4 Report
This manuscript propose an online multiple calibrations spinning LiDAR inertial odometry (OMC-SLIO) method, which can online estimate multiple extrinsic parameters among LiDAR, encoder, and IMU as well as odometry states. Experimental results show that author's OMC-SLIO is effective and attains excellent performance. The study of this manuscript has certain research significance and promotion value, but there are still some areas for further optimization.
1. The introduction and summary of the current research status is not sufficient, and it is suggested to add further and summarize the advantages and disadvantages of previous studies in Section 1.
2. For the introduction of figures and tables in the text, the words shown in Figure XX or Table YY should be given in the text first, and then the corresponding figures or tables should be given, otherwise, it is incorrect.
3. In the formula, the matrix and vector need to be thickened and skewed, and the general variables need to be skewed. Some formulas and variables in this paper are not written according to the standard, so please revise and check the full text.
4. The conclusion in the manuscript is not sufficiently written and needs further improvement and summary.
5. In Figure 3, the superscript of Pi does not select the correct mode in the formula editor, which causes the subscript to be presented in the form of a box and needs to be revised.
Author Response
Dear Editor,
Thank you for kind work of our manuscript (an updated version of previous manuscript: “sensors-2020902 OMC-SLIO: Online Multiple Calibrations Spinning LiDAR Inertial Odometry”), with an opportunity to address the reviewers’ concerns. Based on the comments of the reviewers, we have made corresponding changes noticeable in updated manuscript and answered all concerned questions in this letter.
We are uploading our point-by-point response to the comments (below) and an updated manuscript with changes highlighted.
Best wishes,
Guijin Wang, et al.
Thanks very much for reviewer’s comments. All changes mentioned below are green highlighted in the updated manuscript.
Response to Reviewer 4:
Concern 1: The introduction and summary of the current research status is not sufficient, and it is suggested to add further and summarize the advantages and disadvantages of previous studies in Section 1.
Author response: Thanks for the reviewer’s recommendation. In the updated manuscript, our condensed summary of the relevant research work is presented in the Introduction section, as.
“Therefore, existing self-calibration LIO methods, whether filter-based [10,11] or optimization-based [12,13], do not take the mechanism calibration into account and require the LiDAR to be rigidly attached to the IMU, which cannot be implemented for SLiDAR. In addition, existing mechanism calibration methods regarding the calibration between the rotating mechanism and the LIDAR are offline and have very complex implementation principles that hinder their integration to self-calibration LIO [14,22,23].”
More detailed related work is described and summarized in the Related Works section.
Concern 2: For the introduction of figures and tables in the text, the words shown in Figure XX or Table YY should be given in the text first, and then the corresponding figures or tables should be given, otherwise, it is incorrect.
Author response: Thanks for pointing out the mistakes. In the updated manuscript, we have adjusted all figures or tables and their textual descriptions according to your suggested writing specifications.
Concern 3: In the formula, the matrix and vector need to be thickened and skewed, and the general variables need to be skewed. Some formulas and variables in this paper are not written according to the standard, so please revise and check the full text.
Author response: Thanks for pointing out the mistakes. In the updated manuscript, we have revised and modified the formula and variable as your suggested standard.
Concern 4: The conclusion in the manuscript is not sufficiently written and needs further improvement and summary.
Author response: Thanks for the reviewer’s recommendation. In the updated Conclusion section, we have not only further optimized the summary of our work and analyzed the advantages and disadvantages of our current work. We have also added a Discussion section dedicated to the analysis and discussion of the reasons why our approach could yield better results.
Concern 5: In Figure 3, the superscript of Pi does not select the correct mode in the formula editor, which causes the subscript to be presented in the form of a box and needs to be revised.
Author response: Thanks for pointing out our carelessness. We have corrected this point of error in the updated manuscript.
Round 2
Reviewer 1 Report
The paper is now in better shape. I have no comment and agree to publish.
Reviewer 3 Report
Thanks for your reponse and modification, the paper can be accepted on its current contents.